# A Survey on the Abstraction and Reasoning Corpus

**Severin Bratus**                                     *sbratus@ethz.ch*
**David F. Jenny**                                     *davjenny@ethz.ch*
**Andreas Plesner**                                    *aplesner@ethz.ch*
**Roger Wattenhofer**                                  *wattenhofer@ethz.ch*
*Distributed Computing Group*
*Computer Engineering and Networks Laboratory*
*ETH Zürich*

**Reviewed on OpenReview:** *https://openreview.net/forum?id=qzFxBcK9Cg*

## Abstract

Chollet (2019) proposed a definition of intelligence that emphasizes efficiency in skill acquisition rather than performance on a predefined set of tasks, and introduced the Abstraction and Reasoning Corpus (ARC-v1, or ARC-AGI-1) as a challenge benchmark for machine learning research. In the following years, ARC and the associated competitions have highlighted fundamental limitations of classical deep learning approaches and underscored the need for new ideas in abstract reasoning. This has incentivized extensive trial-and-error exploration, resulting in a wide variety of methods applied to the corpus. As ARC-v2 was released in March 2025, this literature survey provides a systematic breadth-first overview of the methods applied to ARC-v1 in the six years since its introduction, prior to version 2, and covers early developments for ARC-v2 and ARC Prize 2025. We apply a taxonomy distinguishing inductive (which explicitly construct transformation rules) and transductive approaches (which directly map inputs to outputs), examine the ecosystem of enabling techniques and auxiliary datasets, and synthesize patterns, trade-offs, and underexplored areas across the research landscape. Our goal is to provide newcomers with a comprehensive foundation for understanding existing approaches and identifying promising research directions in abstract reasoning.

## 1 Introduction

Generalization in abstraction and reasoning capabilities remains one of the fundamental open problems in artificial intelligence research (Lake et al., 2017; Dziri et al., 2023). While contemporary AI systems have demonstrated impressive capabilities in specialized domains, their ability to adapt to novel situations and generalize with respect to underlying principles still lags behind human performance. This gap is particularly evident in tasks requiring abstract reasoning from limited examples.

To address this challenge, Chollet (2019, p. 27) argued in favor of emphasizing breadth of generalization in acquiring skills, rather than raw skill, as the focus of a working definition of intelligence:

> The intelligence of a system is a measure of its skill-acquisition efficiency [in the information-theoretic sense] over a scope of tasks, with respect to priors, experience, and generalization difficulty.

Based on this definition, Chollet introduced the Abstraction and Reasoning Corpus (ARC), a benchmark designed to measure general intelligence through skill-acquisition efficiency rather than performance on specific, predefined tasks. ARC tests the ability of systems to infer abstract rules from few examples and apply them to new situations, leveraging only the priors common to human cognition, referred to as Core Knowledge.

Since its introduction in 2019, ARC-v1 has catalyzed diverse research directions and approaches. It has been the subject of four major public competitions (in 2020, 2022, 2023, and 2024) with monetary prizes, attracting entries from industry and academic research teams (Li et al., 2024a; Akyürek et al., 2024; Cole & Osman, 2025). These competitions have driven significant methodological innovation, with approaches ranging from purely symbolic program synthesis (Wind, 2020; Xu et al., 2023) to neural architectures with test-time adaptation (Cole & Osman, 2025; Barbadillo, 2024a), and many hybrids in between (Bober-Irizar & Banerjee, 2024; Butt et al., 2024).

Since its initial release in 2019, ARC has evolved through several versions. ARC-v2 was released in March 2025, alongside the ARC Prize 2025 competition launch. Maintaining the original design principles and format, it features increased task complexity and improved calibration methodology (Chollet et al., 2025b). ARC-v3 was released in March 2026 (ARC Prize Foundation, 2026). It introduces agent-environment interaction over time, in the setting of arcade-like games, expanding the scope of the benchmark beyond static transformations.

The present survey aims to provide researchers interested in developing novel approaches for ARC with a broad overview of past efforts prior to ARC-v2, whether successful or not. While several notable blog posts provide insights into specific ARC methods (Hemens, 2025; Barbadillo, 2024b;c), these resources are necessarily selective, offering detailed analysis of particular approaches rather than exhaustive and systematic coverage of the field. A concurrent work by Zinkevich (2025) provides a systematic review of 62 studies on ARC-v1 and ARC-v2, however, it does not offer a detailed exposition of the area. Related work on abstract visual reasoning benchmarks (Malkinski & Mandziuk, 2023) and program synthesis (Gulwani et al., 2017) offers relevant context but addresses different problem scopes. The ARC literature has grown rapidly but remains fragmented across research communities. Methods vary substantially along multiple dimensions—accuracy, computational costs, reliance on Core Knowledge priors, generalization capacity, domain specificity, learning open-endedness, and explainability—making it challenging for newcomers to gain a comprehensive understanding of the full landscape of existing approaches and their relative merits. With ARC-v2 and ARC-v3 recently released, a unifying taxonomy and comprehensive overview of ARC-v1 efforts is timely.

We address this gap through three main contributions: (1) a taxonomy organizing ARC solvers primarily by their core design—inductive methods that explicitly construct transformation rules versus transductive methods that directly map inputs to outputs. While the inductive–transductive distinction itself has been noted in prior work (Li et al., 2024a), we provide explicit problem formulations and a systematic organizational structure with subcategories that capture different rule representations, solver designs, and enabling techniques across the ARC research landscape; (2) a systematic classification and analysis of approximately 100 works according to this taxonomy (see Appendix A for literature collection details), synthesizing their strengths, limitations, and complementary characteristics; and (3) an examination of the broader ARC ecosystem, including human performance studies and auxiliary datasets.

Taken together, these contributions render fragmented research into a coherent body of work with shared terminology and structure, enabling direct comparison between methods typically evaluated in isolation and exposing underexplored research directions.

The remainder of this paper is organized as follows: Section 2 describes the ARC benchmark itself, including its structure and design principles. Section 3 presents a taxonomy of ARC methods and contributions. Section 4 reviews studies on human performance on ARC as a reference point. Sections 5 and 6 examine inductive and transductive approaches respectively. Section 7 reviews the ecosystem of auxiliary datasets. Finally, Section 8 discusses broader implications and future directions.

## 2   The Abstraction and Reasoning Corpus

ARC is a benchmark designed to assess few-shot reasoning and abstraction in AI systems by requiring generalization from limited examples to novel problem instances, distinguishing it from benchmarks that primarily measure learned task-specific knowledge. This section provides an overview of ARC's structure,

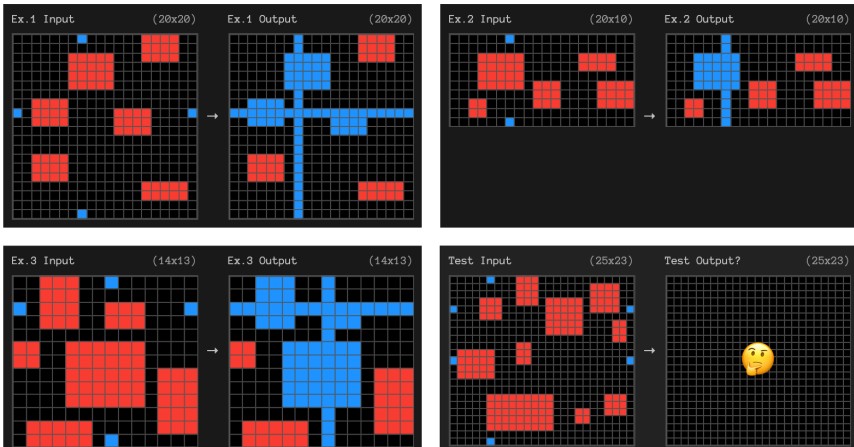

Figure 1: Sample ARC-v1 task `0d87d2a6`. Here the transformation rule may be formulated as follows: Connect opposite blue pixels with horizontal or vertical blue lines, and recolor the rectangles encountered along the path to blue. The solver is expected to correctly infer the test output grid dimensions (in this task the same as the input dimensions), and content (here the test output is marked by an emoji). Source: `arcprize.org`.

design principles, and evolution, establishing the context for our subsequent analysis of methods. We refer the reader to Chollet (2019) and Chollet et al. (2025a) for more details.

ARC consists of a collection of visual reasoning tasks, each requiring the solver to infer an abstract transformation rule from a few examples. Each ARC task includes a few input-output grid pairs (typically 2–4) as examples (the **support set**), along with one or two test inputs (the **query set**) for which the solver must predict the corresponding outputs. The input and output grids are limited to 30-by-30 cells, each containing an integer from 0 to 9 to represent different colors.

Figure 1 shows a sample ARC task, where the transformation involves connecting opposite blue pixels with blue lines and recoloring the rectangles along the way to blue. ARC transformations range from simple geometric operations to complex pattern recognition and relational reasoning, with many concepts appearing in only one or a few tasks. This diversity makes ARC particularly valuable for studying capability acquisition and generalization: even when tasks require different specific capabilities, these can often be abstracted to more general concepts, or as being made up of more general primitives, that transfer across tasks—for instance, the phenomena of objects falling downward ('gravity') and objects moving towards other target objects ('magnetism') can both be viewed as instances of 'attraction.'

ARC may be seen as an example of abductive reasoning (Douven, 2025; Lim et al., 2024), that is, inference to the best explanation for an incomplete set of observations. Admittedly, the notion of which explanation is best is human-centric — in the case of two conflicting possible explanations, the one that seems 'simpler' and more 'natural' should be chosen. It should also be noted that for some tasks the true transformation rule cannot be derived from one (or even two) of the examples — instead *all* examples must be considered at once.

The design of ARC is inspired by Core Knowledge theory (Spelke & Kinzler, 2007), which posits that from an early age humans possess innate knowledge related to objects, goal-directed behavior, numbers, and space (see Figure 2). By restricting the assumed priors to these domains, ARC aims to create a fair comparison between human and machine intelligence that focuses on reasoning capabilities rather than accumulated knowledge.

The original ARC-v1 corpus comprises 1,000 tasks, divided into four subsets: a public training set (400 tasks), a public evaluation set (400 tasks), a private evaluation set (100 tasks), and a semi-private evaluation set (100

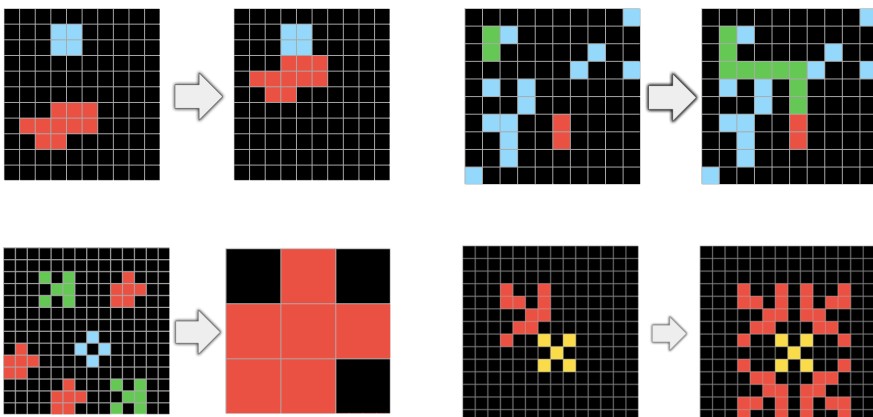

Figure 2: Examples of ARC transformation rules illustrating the Core Knowledge priors. From left to right, top to bottom: recognizing objects and interaction; goal-directed behavior (here as path-finding); numerical reasoning (counting, comparison); and spatial reasoning (symmetry). Adapted from Chollet (2019).

tasks). The private evaluation set is kept unavailable, and used for controlled assessment in competitions, ensuring that systems are evaluated on truly unseen tasks. While the semi-private evaluation set is also hidden, it is used to evaluate methods that rely on closed-source third-party APIs and has a higher risk of data leakage. Note that the difficulty among the task sets in ARC-v1 is not uniformly calibrated: evaluation set tasks are significantly harder than training set tasks, and private evaluation set tasks are thought to be harder than public ones (LeGris et al., 2024; Chollet et al., 2025b).

## 3    A Taxonomy of Methods and Contributions

The ARC research landscape includes computational methods, auxiliary datasets, and tools, human performance studies, and theoretical work on design and intelligence. Figure 3 shows our taxonomy of the ARC research landscape. Following the distinction by Li et al. (2024a), solvers are divided into two designs: inductive methods, which explicitly construct representations of transformation rules, and transductive methods, which map inputs to outputs directly. The inductive methods are further divided based on the type of rule representation they construct: programs (in domain-specific or general-purpose programming languages), natural-language descriptions, or latent vector features. Besides this hierarchy, our survey also highlights enabling techniques and contributions—ARC-specific infrastructure (DSLs, auxiliary datasets) as well as general techniques (such as search algorithms, test-time adaptation).

The transductive–inductive distinction applied in this survey is a generalization of the distinction made by Li et al. (2024a), who consider only certain neural configurations. In this survey the division is not made formal, nor is it fundamental in terms of capabilities, as the transductive framework subsumes the inductive one, and vice versa; the two classes are in theory equally expressive. However, in practice we may approximately distinguish methods that infer a compact reusable rule representation explaining the support data, from those that do not. Further, inductive and transductive designs tend to result in somewhat different generalization behavior. This distinction has implications for explainability, generalization, and the transformation types that the methods may be better suited to capture.

The following subsections describe the categories of the taxonomy (Figure 3) in more detail, establishing the framework that structures our review in later sections.

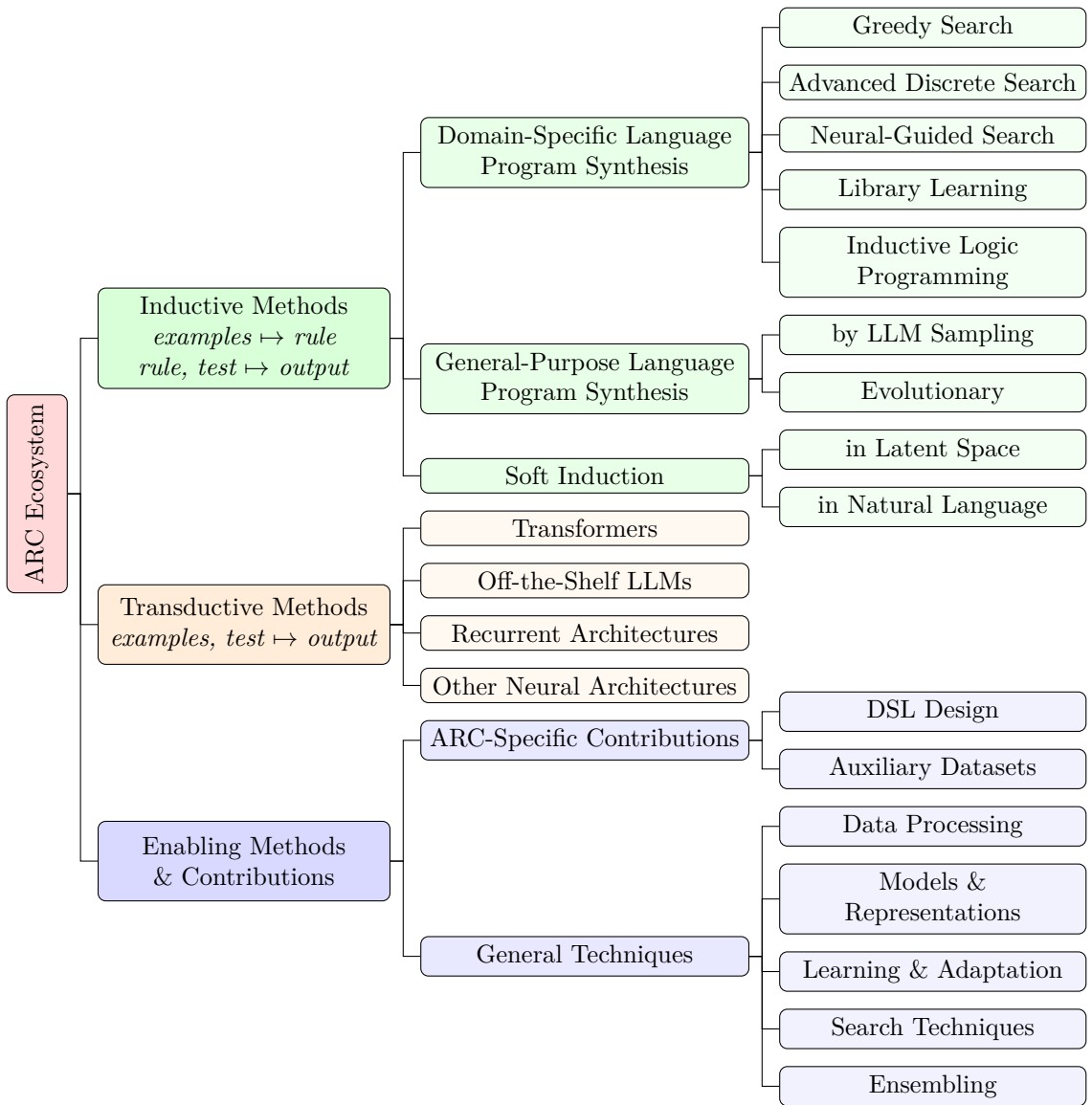

Figure 3: Taxonomy of ARC research contributions, including enabling contributions designed specifically for ARC (DSLs, auxiliary datasets), general techniques borrowed from other fields (search, adaptation, pretrained models), and computational methods. Computational methods are categorized as inductive (explicitly inferring a rule or program from examples before applying it to test inputs) or transductive (directly mapping from examples and test inputs to outputs without an explicit rule representation). Each node is subdivided by whatever criteria are most informative for its members, rather than by a single axis across nodes, and a criterion need not be exclusive to where it appears; for instance, 'LLM Sampling' is placed under 'General-Purpose Language Program Synthesis' but characterizes some methods elsewhere in the tree as well. Leaf nodes group methods that share similar strategies, problems, or considerations and need not exhaustively partition their parents. The taxonomy reflects the state of ARC research at the time of writing, and we expect the taxonomy to warrant new entries or finer distinctions over time.

### 3.1 Inductive Methods

Inductive methods explicitly infer a mapping $f$, often in the form of a program, given the support examples, then apply this mapping to query inputs:

$$\text{Step 1:} \quad \{(x_i, y_i)\}_{i=1}^n \mapsto f$$
$$\text{Step 2:} \quad y_* = f(x_*)$$

Here $(x_i, y_i)$ are the $i$-th example input-output grid pair, and $(x_*, y_*)$ are the query input-output pair.

These methods aim to discover general underlying rules—analogous to writing a Sudoku-solving algorithm versus simply filling in one instance of the puzzle. These methods divide into three main subcategories:

- **DSL-Based Program Synthesis:** searches over programs composed in a domain-specific language (DSL), typically carefully designed by a human. The DSL primitives encode assumptions about the types of transformations likely to appear in ARC tasks. Methods range from brute force enumeration (Wind, 2020) to more sophisticated approaches using heuristic search, neural-guided search, and library learning.

- **General-Purpose Language Program Synthesis:** uses general-purpose programming languages,[1] typically Python, to express transformation rules. This approach trades the structured constraints of DSLs for greater expressiveness and flexibility. Methods in this category are typically guided by large language models (Greenblatt, 2024), with some employing evolutionary search to refine populations of programs or reasoning traces (Berman, 2024; 2025).

- **Soft Induction:** constructs rule representations other than discrete executable programs. This includes methods that learn continuous latent program representations (Bonnet & Macfarlane, 2024; Assouel et al., 2022) or natural-language descriptions of rules (Camposampiero et al., 2023). Although these methods resemble program synthesis, they differ in a key way: the inferred representations are not coupled to a deterministic executor with formal semantics. As a result, the rule may be difficult or impossible to execute predictably or debug.

### 3.2 Transductive Methods

Transductive Methods directly map from the support examples and query input to output, without explicitly representing the underlying rule:

$$(\{(x_i, y_i)\}_{i=1}^n, \ x_*) \mapsto y_*$$

These methods solve single instances through pattern matching to examples—analogous to filling in one instance of a Sudoku puzzle by recognizing similar patterns, without necessarily deriving the general solving procedure.

Transductive methods include transformer models trained to directly predict output grids (Franzen et al., 2024; 2025; Sorokin & Puget, 2025), or off-the-shelf large language models prompted likewise, in text format (Xu et al., 2024). Transformers commonly use test-time fine-tuning to adapt their parameters to the examples, sometimes supported by meta-learning–based pretraining (Mondorf et al., 2025).

### 3.3 Discussion of the Transductive–Inductive Distinction

The distinction between inductive and transductive design has significant implications: neural solvers of comparable capacity have been found to show highly-complementary performance (Li et al., 2024a). Consider the intuitive exposition by Li et al. (2024a, p. 2):

---

[1]Sometimes also called 'open-ended programming languages' (Chollet et al., 2025a).

> Induction captures the notion that a learner should first explain the training data, then use that explanation to make predictions. Inductive learners can perform better by spending more time optimizing or searching for better explanations, using the [support] examples to score candidate functions. Transduction instead captures the intuition that the training examples themselves should play a direct role in generating new predictions, and that successful prediction need not require an explicit explanation.

Arguing from first principles, inductive methods may be better suited to generalize to variations of the transformation rule within one task, since they isolate the underlying rules from the test input. This is similar to scientific modeling, where formulas that capture underlying natural laws may generalize beyond the observed regime. In contrast, a neural network that interpolates patterns in the training data may achieve strong in-distribution accuracy, yet fail under out-of-distribution shifts because it does not isolate the governing rule. If the rule is represented as a program or symbolic structure which is not too large or complex, then the explicit rule may be examined, debugged, and understood by humans. However, these methods face substantial challenges: searching vast program spaces is computationally expensive, and DSLs, despite careful design, may not be able to represent all viable transforms. Typically, inductive methods require predefined program spaces, limiting their flexibility.

Transductive neural methods generally impose fewer restrictions on possible transformations—thus they are well suited to reproduce output patterns recognized in training data, as by the Universal Approximation Theorem (Hornik et al., 1989). However, they may struggle to capture complex compositional transformations that are naturally expressed as sequences of operations. Their learned representations are typically opaque, making it difficult to understand why a particular prediction was made or to verify correctness without execution.

Though neural solvers are generally transductive, there are exceptions that blur the boundary in the transductive–inductive distinction: For example, Bonnet & Macfarlane (2024) introduced the Latent Program Network (LPN), which performs inductive reasoning in a continuous latent space without requiring a predefined DSL—producing "soft" programs that can be optimized by gradient descent at inference time. This approach may be seen as an inductive method without discrete program search (which is uncommon for this category), or as a transductive neural method with architectural modifications to enable an explicitly separated—though implicit in the sense of being distributed—representation of a rule. Even if underperforming presently, boundary-crossing methods open promising research directions by addressing longstanding limitations rather than refining established paradigms, and therefore may have more potential for further growth.

An alternative taxonomy of methods would divide the methods broadly into symbolic, neural, and hybrid neurosymbolic, based on the type of the primary underlying representations used in the method. However, we found that it is more informative to classify methods based on the problem framing; inductive methods aim to first infer the rule, which then can be applied to the test input, whereas transductive methods aim to predict the test output directly from the examples. Technically, the distinction is in whether or not a representation of the hypothesis transformation rule is separately derived from the support examples. Inductive methods are mostly symbolic or neurosymbolic, whereas direct neural methods are transductive. Even if a neural network, with sufficient capacity and training, might obtain an internal rule representation, which is approximately invariant to input grid variations (challenging as it is), the representations being distributed, the rule representation would not be readily separable. Understanding when neural architectures develop internal rule representations, and whether such representations are functionally equivalent to explicit induced programs, remains an important open question extending beyond ARC.

### 3.4 Enabling Methods and Contributions

Enabling methods and contributions encompass both ARC-specific contributions and general techniques borrowed from other fields, forming the foundational infrastructure that computational methods build upon.

ARC-specific contributions include custom domain-specific languages (DSLs)[2] (Hodel, 2024b; Wind, 2020; Bober-Irizar & Banerjee, 2024; Ainooson et al., 2023) designed to express transformation rules efficiently, and auxiliary datasets that extend or complement the original ARC corpus (see Section 7 for details). General techniques include methods imported from broader machine learning and AI research without substantial ARC-specific adaptation (see Figure 3). We group them into five categories and discuss them and their important members (bold) below:

- **Data Processing:** includes strategies that range from simple **Augmentation** transformations (rotations, reflections) (Hodel, 2024a) to more sophisticated **Synthesis** of tasks, such as synthetic task generation by LLMs (Li et al., 2024a), or training-time data generation phases (Ellis et al., 2020). These have been widely adopted to address the problem of limited training data available in ARC. Some approaches even augment data at inference time (Franzen et al., 2024). Some methods blur this boundary by generating their own learning experiences, e.g., automatic curriculum generation, or DreamCoder's (Ellis et al., 2020) generation of synthetic training problems ('fantasies') during its 'dreaming' phase. The relevant mechanisms can be seen as loosely contributing to this category as well, depending on their generality.

- **Models & Representations:** covers both the models brought to bear on ARC (including their underlying architectures, not only their pretrained weights) and the representational choices these models rely on. **Pretrained (V)LMs**, large (vision-)language models pretrained on vast amounts of data, have various applications in the ARC domain, including in chain-of-thought, fine-tuned models (Cole & Osman, 2025), and program generation (Greenblatt, 2024). On the representation side, several works adopted **Graph Representations**, representing ARC transformations as operations on graph structures rather than directly on grids, hypothesizing that object-centric graph representations better capture the relational reasoning present in many ARC tasks (Xu et al., 2023; Neumann & Pintér, 2023; Lim et al., 2024; Lei et al., 2024). These methods indeed tended to show better performance on object-centric tasks (35–50% vs 15–20%), supporting the hypothesis. GNNs have been applied in ARC to human error modeling (Do et al., 2024) and relation classification (Zeng et al., 2022).

- **Learning & Adaptation:** covers how solvers are trained and how they adapt to individual tasks, spanning both training-time learning paradigms and inference-time adaptation. **Reinforcement Learning** provides frameworks, insights, and algorithms that have been adapted to ARC, treating task-solving as a sequential decision-making problem. Chollet et al. (2025a) expect specialist neurosymbolic systems like AlphaProof (Hubert et al., 2025) to perform well on ARC, by making neural models guide branching decisions of discrete program search processes (probably with use of RL). **Test-Time Adaptation (TTA)** is the broad concept of adapting to the new data provided at inference time. Notably, Test-Time Training (TTT) has emerged as a breakthrough technique in transductive methods (Akyürek et al., 2024), allowing models to adapt to the provided support examples by adjusting their parameters at inference time. We discuss TTT specifically in Section 6.2.

- **Search Techniques:** encompass various algorithms for exploring solution spaces, from beam search and Monte Carlo Tree Search (MCTS) to evolutionary algorithms. These are most essential for methods with well-defined search spaces (e.g., program synthesis over DSLs). LLM decoding may also be viewed as a search process, as exemplified by applications of depth-first search (Franzen et al., 2024) and MCTS (Inoue et al., 2025).

- **Ensembling:** is the process of combining several approaches to generate an improved final output. Multiple studies report accuracy gains by ensembling neural and symbolic solvers (Akyürek et al., 2024; Barbadillo, 2024a), suggesting the two are somewhat complementary in the ARC domain (at a small-to-medium scale of compute). In particular, Li et al. (2024a) found that most problems solved by their inductive neurosymbolic transformer-based solver were not solved by the transductive neural one, and vice versa.

---

[2]The design of DSLs involves numerous theoretical and practical considerations, including expressivity, bias, and amenability to search. However, a detailed treatment is beyond the scope of this survey. We refer readers to the cited works for in-depth discussions.

## 4   Human Performance

Observing human performance on ARC provides context for evaluating AI methods and identifying the cognitive capabilities that remain challenging to automate. Humans do not compare fairly to minimal-prior ARC solvers, as they bring extensive prior knowledge beyond the core priors ARC intends to measure.

Human problem solving does not appear to clearly fall into either side of the inductive–transductive distinction. Dual-process theories of human reasoning (Evans & Stanovich, 2013) propose a distinction between fast, intuitive responses, and slower, deliberative reasoning, which loosely parallels transductive and inductive approaches, respectively. For simple pattern-oriented tasks, the human solvers seem to find a solution easily and intuitively, akin to transductive neural methods, without formulating the rule in an aware manner; whereas for complex, non-obvious, or procedural tasks, a human solver might have to attentively consider and check multiple hypothetical rules before arriving at the right one, formulating a high-level algorithmic representation thereof, possibly in a form of mental language hypothesized by Fodor (1975).

In what follows, we review empirical findings on human performance, error patterns, and cognitive processes when solving ARC tasks.

**Accuracy estimates**   Average human performance on ARC-v1 was estimated to be 76.2% on the training set, and 64.2% on the evaluation set by a large-scale study with $n = 1,729$ crowd-workers (LeGris et al., 2024). However, previous small-scale studies on a subset of tasks reported accuracies above 80% (Johnson et al., 2021; Acquaviva et al., 2022).

**Error analysis**   Humans and LLMs (directly prompted) differ in the types of errors they make. Humans make errors, such as missing a few pixels, while grasping the correct concept (Johnson et al., 2021; Moskvichev et al., 2023). LLMs, akin to children in the pre-analogical stage, will attempt to copy and combine grids seen previously (Opiełka et al., 2024). Humans are also substantially better at self-correction from minimal feedback, and can flexibly consider alternative solutions (LeGris et al., 2024). Results of Li et al. (2024a) suggested that humans struggle with some tasks that are simple when represented in code, and thus comparatively easy for a program synthesis system.

**Cognitive processes & verbalization**   When describing how to solve an ARC task, human participants communicate 'natural programs' that contain a wide range of primitives, as well as ample 'non-executable information' in the form of comments and clarifications on framing or validation (Acquaviva et al., 2022). Johnson et al. (2021) proposed that humans generate hypotheses based on background knowledge much beyond a limited set of primitives, and have a flexible context-dependent notion of objects. Moskvichev et al. (2023) suggested that inductive priors relevant to ARC are represented at the level of the human visual system.

## 5   Inductive Methods

Inductive methods approach ARC by explicitly representing transformation rules, typically as programs that capture the underlying pattern demonstrated in the support examples. As introduced in Section 3.1, these methods follow a two-step process: first inferring a general mapping $f$ from the support set, then applying this mapping to test inputs.

**Advantages**   The resulting representations may be inspected or debugged by humans, supporting interpretability (Ellis et al., 2020; Ferré, 2024) and the explicit capture of transformation logic may support better generalization within one task, across input variations—in the sense that a robust rule representation will perform correctly even for out-of-distribution inputs (Li et al., 2024a; Bonnet & Macfarlane, 2024). For example, augmentations such as altering colors in the input grid might mislead a transductive method (Franzen et al., 2024; Hu et al., 2025), where an inductive method might have found a color-agnostic rule.

**Challenges**   The space of possible programs grows combinatorially with program length and the size of the primitive set, making systematic enumeration intractable even with advanced search strategies

(Gulwani et al., 2017). Another problem, more conceptual than practical, is that of selecting the *true* intended generalizable program among all those functionally-distinct programs that satisfy the support examples. In ARC, the true program corresponds to the one that obeys the principle of parsimony, Occam's razor, in a modified human-centric perceptual–predictive sense, [3] meaning the true program represents the simplest possible explanation of the observations (the support data), as perceived by a human—and is therefore robust to future uncertainty. Yet in practice, it is challenging enough to find a single non-trivial program fitting the examples (Ainooson et al., 2023), and thus the selection problem is often moot in practice. Further, DSLs are designed in a way such that probable solutions are easier to express and therefore more probable to be found in search (Wind, 2020).

The form of the rule representation—whether in a domain-specific language, general-purpose code, or continuous latent space—affects the type of transformations that can be truthfully expressed, how efficiently they can be discovered, and crucially, the optimization biases in the search for programs likely to generalize.

In most inductive approaches, the Core Knowledge priors are represented by human design in explicit symbolic constructs, such as the DSL primitives (Wind, 2020; Neumann & Pintér, 2023; Ferré, 2024; Ainooson et al., 2023; Acquaviva et al., 2022), construction of graphs (Xu et al., 2023; Lim et al., 2024), or grammars (Witt et al., 2023). This introduces substantial human bias into the search process, and can severely limit the range of tasks that generalize well. For instance, the exemplary DSL solver Icecuber (§5.1.1) scored 17% on the ARC-v1 semi-private evaluation set but only 1.6% on ARC-v2 (Chollet et al., 2025b).

Inductive approaches have produced some notable successes on ARC, particularly when the transformation aligns well with the primitives or structure of the chosen representation. In this section, we examine three major categories of inductive methods: DSL-based program synthesis (Section 5.1), which searches over human-designed domain-specific languages; general-purpose language program synthesis (Section 5.2), typically guided by large language models; and soft induction (Section 5.3), which constructs implicit rule representations in continuous or natural language spaces.

## 5.1 Program Synthesis in Domain-Specific Languages

DSL-based program synthesis approaches search for programs within domain-specific languages, where the primitives and composition rules encode assumptions about the types of transformations likely to appear in ARC tasks. The design of the DSL is critical: it determines which transformations can be expressed, how concisely they can be represented, and which programs are prioritized during search.

### 5.1.1 Greedy Search

The 2020 ARC-v1 competition winner, 'Icecuber' by Wind (2020), demonstrated the potential of even simple DSL-based approaches through greedy enumerative search. Each candidate program applied up to 4 of 142 possible unary transformations derived from 42 base functions, greedily stacking operations to match the support examples. Despite its simplicity, this approach achieved 20% accuracy on the private evaluation set— a result that remained competitive for years and led to Icecuber being frequently employed in ensembles alongside neural solvers in subsequent competitions. Another notable solver in this category, by 'alijs', scored 40% on the private evaluation set in the 2024 ARC Prize competition (Chollet et al., 2025a).

### 5.1.2 Advanced Discrete Search

Beyond brute-force DSL search, researchers explored a variety of approaches that guide exploration toward promising candidate programs by heuristics or advanced search strategies. These include: multi-level search (Ainooson et al., 2023), grammatical evolution (Fischer et al., 2020), Monte Carlo tree search (Basaldúa, 2020), minimum description length modeling (Ferré, 2024), non-procedural program induction (Neumann & Pintér, 2023), generalized planning (Lei et al., 2024), graph representations, constraint-guided search (Xu et al., 2023; Lim et al., 2024), learning classifier systems (Coombe et al., 2024), and vector symbolic algebras

---

[3]Chollet (2019, pp. 35–36) discusses the relation of Occam's razor to generalization: "generalization describes the capability to deal with future uncertainty, not the capability to compress the behavior that would have been optimal in the past".

(Joffe & Eliasmith, 2025). Many of the above approaches emphasize object-centric representations (Xu et al., 2023; Neumann & Pintér, 2023; Ferré, 2024), which have proved valuable for reducing the search space.

The advanced symbolic methods listed above have generally struggled to scale: most achieved below 20% accuracy on the evaluation set, with the best reporting only slightly above 20% on the training set (Ainooson et al., 2023; Ferré, 2024). Some methods have reported significantly higher accuracies (35–50%) on an object-centric subset of ARC (Xu et al., 2023; Lei et al., 2024), suggesting these methods may be better suited for tasks with clear object-centric relations.

### 5.1.3 Neural-Guided Search

Neural-guided search methods employ neural networks (including LLMs) to learn functions that guide exploration of the DSL program space. They are in part inspired by the success of neural-guidance systems such as AlphaZero (Silver et al., 2018), where neural networks learn to evaluate states and guide tree search.

Several works have explored neural guidance for program synthesis on ARC. Alford et al. (2021) developed bidirectional program search with inverse semantics, guided by reinforcement learning. Batorski et al. (2025) applied transformer-guided combinatorial search. Bednarek & Krawiec (2024) combined reinforcement learning with supervised learning on synthetic tasks, to guide program generation. Butt et al. (2024) trained models on predicted programs and their input-output examples. Adding language-pretraining, LLM-guided DSL synthesis (Singhal & Shroff, 2024; Ouellette, 2024) can outperform both direct LLM prompting as well as common symbolic methods (Barke et al., 2024).

The top neural-guidance methods achieved 15–20% accuracy on the public evaluation set (Butt et al., 2024; Batorski et al., 2025). Their modest performance may suggest current methods still fail to balance learned heuristics with structured search, or that the underlying learning systems (including RL and LLMs) are yet insufficient to learn effective guidance policies.

### 5.1.4 Library Learning

Library learning automatically discovers and abstracts reusable program components from solved tasks, building up a library of higher-level primitives that can simplify future program synthesis. In a successful application of library learning, only a small set of primitives (with sufficient compositional expressiveness) needs to be provided, thus reducing immediate human bias in the domain-specific library design. By learning useful higher-level abstractions, the program length required to express solutions is reduced, mitigating the combinatorial explosion of the search space.

An influential framework in this category is DreamCoder (Ellis et al., 2020), a general "wake-sleep Bayesian program induction" system that learns to synthesize programs for well-defined problems. In the *wake* phase, DreamCoder uses a recognition model (a neural network) and a generative model (determined by a library of primitives) to search for programs that solve training tasks. The *sleep* phase consists of two subphases. In the *abstraction* subphase, the system extends the library of primitives by compressing programs found during the *wake* phase. In the *dreaming* subphase, the recognition network is trained to predict programs for both real and synthetic tasks.

While DreamCoder's approach of learning compositional abstractions seemed conceptually well-suited for ARC's diverse task space, direct applications struggled to scale to ARC's complexity (Alford et al., 2021; Peter, 2022). The introduction of an adapted custom DSL and recognition model resulted in a significant improvement, however still yielding only 5% on the public evaluation set (Bober-Irizar & Banerjee, 2024).

The limited success of library learning on ARC may be explained in a few ways: the diversity of tasks hinders the discovery of reusable abstractions, the visual–spatial nature of ARC may demand abstraction mechanisms different from symbolic domains, or the approach may require greater scale and computational resources than explored so far. Nevertheless, library learning had been a source of inspiration for methods in Evolutionary General-Purpose Language Program Synthesis (see §5.2.2), leading to significant results.

### 5.1.5 Inductive Logic Programming

Inductive Logic Programming (ILP) represents a form of logic-based program synthesis that learns explainable rules from few examples (Cropper & Dumančić, 2022). ILP's focus on interpretable rules and its ability to learn from minimal data make it conceptually appealing for ARC. However, ILP methods struggle to scale to the vast hypothesis space of 2D ARC tasks.

Recognizing this, several works evaluated ILP systems on the simpler 1D-ARC corpus (Xu et al., 2024), where they achieved up to 90% accuracy (Cropper & Cerna, 2025; Hocquette et al., 2024). On the original 2D ARC, progress stemmed from structural innovations: relational decomposition (Hocquette & Cropper, 2025) and divide-and-conquer strategies (Witt et al., 2023) improved training set accuracy from near-zero to 20%.

## 5.2 Program Synthesis in General-Purpose Languages

General-purpose language program synthesis approaches use general-purpose programming languages to express transformation rules, trading the structured constraints of domain-specific languages for greater expressiveness and flexibility.

In DSL-based approaches, the primitive operations and composition rules are predefined to make search tractable: individual steps are well-defined atomic operations, enabling even enumerative search to explore valid programs. In a general-purpose programming language, no such constraints are imposed, thus requiring a different methodology.

Without curated DSLs, these methods depend heavily on large language models pretrained on code to guide program generation —typically in Python with natural-language comments. LLMs' priors of common programming patterns, algorithms, and problem-solving strategies serve as a useful bias toward generalizable programs.

### 5.2.1 LLM Sampling

Notably, Greenblatt (2024) achieved 50% on the public evaluation set by using GPT-4o to sample a large number (8,000) of Python programs per task, selecting the most likely correct ones, and then iteratively asking GPT-4o to correct them. The method relied heavily on few-shot prompting with detailed step-by-step reasoning, specialized prompts for different types of grid transformations (fixed or changing size), and engineered textual representations of input-output grids. Ensembling multiple prompt variants coupled with post-generation revision leads to 50% accuracy on the public evaluation set.

OpenAI's o3-based entry shortly after ARC Prize 2024 competition achieved a breakthrough, with human-comparable accuracies, in the range of 75–90%, though spending around $200 in compute per task. The exact nature of the solution remains unknown. It is most probable that the high compute per task is due to test-time search, which may be over programs, or over natural-language chains-of-thought. (Chollet, 2024)

### 5.2.2 Evolutionary Program Synthesis and Test-Time Compute

Evolutionary program synthesis methods iteratively apply selection and variation over populations of LLM-generated programs. Initially drawing inspiration from the success of massive sampling of Python programs (Greenblatt, 2024), they introduce more structure to the population refinement cycle.

Berman (2024) pioneered evolutionary test-time compute for ARC, achieving 53.6% on the evaluation set. An LLM is prompted to generate Python functions for transforms; the best-scoring functions are then used in new prompts. The scores are computed based on how well a function predicts the support outputs. This process is repeated multiple times, ultimately generating up to 500 functions using 31 dynamic prompts per challenge. While effective, this approach treats each task independently, failing to transfer learned knowledge across tasks.

Pang (2025) addressed this by adopting principles of library learning (Ellis et al., 2020): their improvement is to keep a library of best-scoring functions, thus allowing some concepts applied earlier to be re-applied

later. This system achieved 77.1% on ARC-v1 and 26.0% on ARC-v2 using only 10 LLM calls per task, highlighting the value of compositional reuse.

Further, Berman (2025) extended his method by replacing Python code with natural-language instructions, achieving 79.6% on ARC-v1 and 29.4% on ARC-v2. The results suggest that natural language can better capture complex transformations than executable code, with an improvement in cost-effectiveness.

A more advanced configuration, SOAR (Pourcel et al., 2025), introduced continual self-improvement of evolutionary search coupled with fine-tuning of the underlying LLM. This dual mutual improvement helped overcome performance plateaus with respect to scaling search compute budget.

### 5.3 Soft Induction

Some methods, while inductive, represent the transformation rules not as explicit, discrete executable programs, but rather as natural-language descriptions, or latent features in continuous space, allowing the use of LLM priors, or gradient-based optimization, respectively. These approaches maintain the inductive frame—first isolating a transferable rule and then applying it—while avoiding the combinatorial challenges of discrete program search. We call this category of methods 'Soft Induction'.

#### 5.3.1 Latent Space Representations

The Latent Program Network (LPN) exemplifies soft induction in continuous latent space (Bonnet & Macfarlane, 2024). LPN is a transformer-based architecture with built-in test-time search that learns a continuous latent space of implicit 'programs'. LPN is composed of three components, trained together end-to-end: (1) a neural Encoder that maps an input-output pair to a distribution in the latent space, representing a range of possible programs that could implement the input-output transformation, (2) a neural Decoder that maps a latent program and an input to a distribution in the output space, (3) an optimization process in the latent space, that iteratively refines the encoder's initial prediction to better fit the support data. Thus, here a transformer effectively serves as an interpreter for latent programs. Once a suitable latent program is identified through optimization, it can be applied to query inputs via the decoder, without access to the encoder or the support examples.

The continuous space allows efficient gradient-based search over programs. However, it is unclear if this use of continuous vectors is sufficient to represent fundamentally discrete grid transformations, which may be compositional or contain many operations. Also, the implicit nature of the latent programs sacrifices the interpretability of traditional program synthesis.

Assouel et al. (2022) explored similar ideas with object-centric neural programs, where transformers act as neural 'executors'. Thoms et al. (2023) embedded input and output grids by a variational autoencoder (VAE), and applying learned transformations as vector arithmetic in the latent space, with limited results.

#### 5.3.2 Natural-Language Representations

While natural language is ambiguous and cannot be deterministically executed like program code, it provides a rich medium convenient for humans and LLMs to interpret and execute. A dataset of such 'natural-language programs' was produced by Acquaviva et al. (2022), as instructions by human participants for other participants to solve ARC tasks, relying on language alone, without access to examples.

Several works have employed natural-language to represent transformations. For instance, Wang et al. (2024) prompt an LLM to propose multiple abstract hypotheses about the problem in natural language, and then to implement the hypotheses as concrete Python programs. Camposampiero et al. (2023) proposed a general framework for solving ARC centered on transforming tasks from the visual domain into natural language. Berman (2025) found that replacing Python with English in an evolutionary program synthesis system improved performance significantly. Ho et al. (2025) explored two styles of descriptions: 'open-ended' and 'program-synthesis', the latter being more structured and specific. The latter resulted in slightly better accuracy.

'Reasoning' language models that use chains-of-thought may be considered to fall under this category, as a reasoning trace may contain some form of re-usable description of the transformation rule, even if the LLM is aiming to ultimately produce an output grid after the reasoning process (see also §6.3).

# 6 Transductive Methods

Transductive methods map directly from support examples and test inputs to outputs, without explicitly constructing an intermediate representation of the transformation rule. As introduced in Section 3.2, these methods tend to solve individual task instances through pattern matching and learned transformations rather than deriving general procedures.

**Advantages** The transductive framing imposes fewer restrictions on representable transformations (any pattern learnable from data can potentially be captured), avoids the combinatorial search challenges of program synthesis, can leverage powerful pattern recognition capabilities of neural networks, and is less reliant on human-designed frameworks (Hornik et al., 1989; Li et al., 2024a; Bonnet & Macfarlane, 2024).

**Challenges** Without an explicit rule representation, these methods must rely entirely on learned patterns to generalize from support examples to test inputs (Mitchell et al., 2023; Bober-Irizar & Banerjee, 2024). This can be particularly difficult when transformations are compositional or require systematic reasoning about multiple interacting operations (Mondorf et al., 2025). Transductive methods may struggle when test inputs differ significantly from support examples in ways that require understanding the underlying rule rather than pattern matching (Mitchell et al., 2023). Additionally, the learned representations are typically opaque—understanding why a particular prediction was made requires examining activation patterns rather than reading an interpretable program.

Despite the challenges, transductive approaches have achieved some of the highest accuracies on ARC (Sorokin & Puget, 2025; Cole & Osman, 2025), particularly when combined with ample data and compute, as well as test-time adaptation techniques that allow models to adjust to individual tasks—typically test-time fine-tuning (Franzen et al., 2025) or recurrent processing (Wang et al., 2025a).

## 6.1 Transformer-based Transduction

Transformer-based transduction is a popular and competitive approach to ARC, in which transformers learn to predict the query output grid, given the query input grid and the support examples. To be processed by transformers, ARC grids are serialized as sequences of tokens, one token per grid pixel.

Transformers may be well-applicable for ARC for multiple reasons. The transformer attention mechanism is able to model long-range relationships to some extent, though hindered by the grids' two-dimensional nature. Large-scale transformers have also shown strong few-shot in-context learning ability (Brown et al., 2020), which makes them attractive for the ARC setting.

However, transformers have multiple significant limitations relevant to ARC. Plain transformers do not impose built-in spatial biases, of the kind that are well-represented in CNNs, and the advantage of transformers over CNNs becomes apparent only when they are trained over a large dataset (Zimerman & Wolf, 2024). On ARC, vision transformers have shown an inability to capture relevant spatial relationships and grid boundaries (Li et al., 2024b). An important factor in transformer limitations in the ARC domain is the lack of large amounts of diverse and high-quality training data, an issue which is only partially addressed by synthetic data generation methods such as RE-ARC (Hodel, 2024a) and Bootstrapping-ARC (Li et al., 2024a).

Transformer variants and architectural modifications proposed in the context of ARC include: an object-centric decision transformer (Park et al., 2023), a 2D normalized transformer (Puget, 2024), lattice symmetry priors via CNN-based learnable soft-attention masks (Atzeni et al., 2023), custom tokenization scheme, object-based positional encoding (Li et al., 2024b), and 2D positional encoding (Costa et al., 2025).

One of the most notable top-performing methods in ARC-v1 was Omni-ARC (Barbadillo, 2024a). Omni-ARC fine-tunes a 500M-parameter transformer to a variety of ARC-related objectives beyond the original objective of input-to-output prediction, such as generating new inputs, verifying outputs, and selecting correct solutions. By reusing the limited training data for multiple objectives, Omni-ARC attempts to build a robust representation of the ARC problem space.

Notably, Hu et al. (2025) approached ARC as a vision problem, specifically image-to-image prediction, where an image is a canvas of sufficient size to include any input or output grid. The authors apply vanilla vision transformers with test-time training, and multi-view inference, attaining 54.5% accuracy on the public evaluation set, trained from scratch. Ablations emphasize the importance of implementing visual priors, through 2D positional embeddings, and a patch tokenization scheme, which significantly augments the data space.

Though some successful approaches use pre-trained LLMs as the base for fine-tuning (Cole & Osman, 2025; Barbadillo, 2024a), it is not yet clear whether or not this helps, as significant results (47% on a subset of tasks) were reported for a relatively smaller model not initialized with language model weights (Fletcher-Hill, 2024). Note that non-LLM transformer-based solvers do not use chain-of-thought, as they reduce the token vocabulary to only a small set of tokens required to represent an ARC grid, thus restricting the potential effect on natural-language pretraining. In contrast, significant improvements were observed from meta-learning for compositionality (Mondorf et al., 2025).

## 6.2 Test-Time Training

Test-time training (TTT, also known as test-time fine-tuning, TTFT) is a mechanism to improve a model's inference-time capabilities by temporarily updating learned model weights using a loss derived from the input data, in the case of ARC, fitting on the support set and its augmentations. The technique itself is not new, and has been known in the literature as local learning (Cleveland, 1979; Atkeson et al., 1997; Bottou & Vapnik, 1992). It was first applied to ARC by Cole & Osman in 2023, resulting in significant success, and since then was adopted by many competition entries which became state-of-the-art (Franzen et al., 2024; Barbadillo, 2024a).

In the context of ARC, the few support set examples are not enough for stochastic gradient descent to be effective, so synthetic data corresponding to the task is generated. Akyürek et al. (2024) identified a few key components of well-performing TTT with respect to ARC:

1. Base fine-tuning on tasks similar to those encountered at test-time

2. Augmented 'leave-one-out' task generation to construct the test-time fine-tuning dataset,

3. Training of task-specific adapters (e.g., LoRA)

4. Self-consistency, or voting over invertible augmentations (e.g., rotations)

Combining the above components Akyürek et al. (2024) achieved state-of-the-art results among research works of this category at the time, with 53% accuracy on the public evaluation set (61.0% ensembled with program synthesis), noting that ablating any of the above components resulted in lower performance.

Performance-wise, test-time training improves accuracy on ARC tasks considerably, up to sixfold (Akyürek et al., 2024). TTT can significantly improve transformers' performance even for small, non-pretrained transformers (Fletcher-Hill, 2024; Zhu et al., 2024). The most successful approaches at the ARC Prize 2024 used TTT, coupled with voting based on augmented variants of the target task (Barbadillo, 2024a; Franzen et al., 2024; Cole & Osman, 2025; Sorokin & Puget, 2025). This includes the top-scoring solution by Cole & Osman (2025), which achieved 58% on the private evaluation set.

A relevant comparison is the Latent Program Network (see §5.3.1), which, like TTT, performs gradient-based adaptation at inference time, but updates only a latent feature rather than the network parameters. Another connection is to neural meta-learning (Hospedales et al., 2022). Meta-learning pretraining is particularly

well suited for TTT (Li et al., 2024a; Mondorf et al., 2025), as it prepares the weight space for few-shot gradient adaptation.

The success of TTT in ARC shows that modern neural networks possess sufficient representational capacity to express many ARC transformations; a primary challenge is identifying the correct transformation from a few examples. By adapting parameters to fit support examples, TTT functions vaguely like program search, but in weight space; restricting the space of possible transformations to those that best explain the evidence.

### 6.3 Off-the-Shelf Large Language Models

Since 2022, large-scale transformer-based language models have been applied to ARC, often focusing on assessing their reasoning capabilities rather than on designing an ARC solver per se (Mirchandani et al., 2023; Mitchell et al., 2023; Lee et al., 2024c). In this section, we focus on transductive applications of non-fine-tuned LLMs, in which LLMs are prompted to solve an ARC task directly, by processing and predicting grids of tokens, and without explicit construction of an intermediate program. LLMs for program synthesis are discussed in Section 5.2.1.

LLMs are transformers with (1) massively scaled capacity, and (2) pretraining on internet text data. Thus, when prompted with an ARC grid serialized as a token sequence, an LLM might rely on priors obtained from observing inherently two-dimensional structures, like tables or ASCII art from its training data, which is mostly 'one-dimensional' text. Another hypothetical advantage given by LLM pretraining for ARC might be in priming some abductive reasoning abilities on high-level representations, from examples of non-spatial natural-language reasoning found on the internet.

Evaluations have shown that when prompted directly, even zero-shot, some GPT models (series 3.5 and 4), though far below human level, can solve a non-trivial number of tasks, often complementary to those solved by symbolic methods (Bober-Irizar & Banerjee, 2024; Mirchandani et al., 2023).

Finetuning of LLMs has been explored with varying outcomes. Finetuning on a large amount of augmented ARC tasks can yield up to a three-fold improvement in evaluation set accuracy (Bikov et al., 2025). However, fine-tuning on basic atomic operations of ARC does not lead LLMs to compositionally generalize to solve full-fledged ARC tasks (Wu et al., 2025).

The choice of representations that LLMs process has a significant effect on their performance. Some works emphasized representing input and output grids as natural-language textual descriptions, suitable for processing by LLMs, which allows even smaller-scale LMs to solve a non-trivial number of tasks (Camposampiero et al., 2023; Qiu et al., 2024; Wang et al., 2024; Glaus, 2024). LLMs tend to struggle to maintain object cohesion when objects are not consecutively distributed in text, whereas object-centric representations enable near-perfect performance of GPT-4 on 1D-ARC tasks (Xu et al., 2024).

Surprisingly, GPT-4v performed worse than GPT-4. Despite the visual nature of ARC, image inputs instead of text resulted in lower accuracy (Mitchell et al., 2023; Palmarini & Mitchell, 2024; Singh et al., 2023).

Other works explored minor interventions, e.g., finding that LLMs performed better when ARC is reframed into a multiple-choice question setting (Shin et al., 2024). Chain-of-thought (CoT) prompting was not shown to have a significant effect in small-scale studies (Galanti & Baron, 2024; Min, 2023).

As of early 2026, applications of chain-of-thought prompting with LLMs such as Gemini 3 Flash, GPT 5.2, and Grok 4 occupy the accuracy–cost Pareto frontier,[4] scoring as high as 90% at $10 per task, in the high-compute range. However, there is evidence of the ARC JSON format being present in some LLMs' training corpora (Knoop, 2025; Mitchell, 2025). Hence, although the tasks in the private evaluation set remain unknown to LLM solvers, the ARC domain itself is no longer novel: LLMs are now ARC-aware, whether through exposure to internet data or targeted engineering.

---

[4]https://arcprize.org/leaderboard

## 6.4 Recurrent Architectures

A novel line of work in ARC is represented by recurrent architectures that iteratively refine their latent features representing the task solution. It is motivated by the limitations of CoT, and in part by the position that "language is primarily a tool for communication rather than thought" (Fedorenko et al., 2024).

Wang et al. (2025a) proposed the Hierarchical Reasoning Model (HRM), with two key features: (1) recurrent hierarchical processing with two networks at distinct frequencies, inspired by fast–slow waves in the brain, and (2) deep supervision (Wang et al., 2015), with multiple supervision steps (loss-gradient-updates) and re-using the latent feature values for initialization in later training loop iterations. Ablations showed that deep supervision was a significant driver of HRM's performance—HRM, trained only on the official ARC dataset with only 27M parameters achieved 32% on the private evaluation set. Building on this work, Jolicoeur-Martineau (2025) simplified and optimized the recurrent setup of HRM, introducing the Tiny Recursive Model (TRM) which attained 40% on the private evaluation set. Test-time fine-tuning of TRM was explored by McGovern (2025), Sorokin & Puget (2025).

## 6.5 Other Neural Architectures

While not a recurrent architecture per se, neural cellular automata (NCA) constitute another form of iterative refinement, and have been applied to ARC with encouraging results, with no ARC-pretraining (Guichard et al., 2025; Xu & Miikkulainen, 2025). Video diffusion models (VDMs) showed higher data efficiency than comparable language-pretrained models, indicating video pretraining may be better suited for ARC (Acuaviva et al., 2025).

CompressARC is a notable solver derived from the minimal description length principle: it solved 20% of the public evaluation set tasks, with only 76K parameters and no pretraining on ARC tasks (Liao & Gu, 2025). It operates by approximating a particular compression algorithm that represents an ARC task (the support and query sets) by the shortest program that reproduces it, along with solutions. CompressARC fits a residual neural network that is equivariant to valid reversible operations, like rotations, reflections, and permutations of support set order or colors. The (test-time) training objective of the network itself is to learn to convert a random tensor into a valid ARC task.

## 6.6 Reinforcement Learning

Reinforcement learning (RL) formulation of ARC attempts to capture the sequential and constructive nature of solving an ARC task, and represents a viable framework that could enable interactive learning from experience.

Lee et al. (2024a) introduced ARCLE, an RL environment, where a state includes the grids, a clipboard, and auxiliary object-centric states. An action consists of the edit operation, and a binary mask over grid pixels. The key challenges in ARCLE are due to (1) high dimensionality of the action space, (2) sparse and delayed rewards, and (3) necessity for few-shot multi-task generalization. The authors applied proximal policy optimization with auxiliary losses, with highly limited results. Diffusion-based Q-learning and model-based RL have yielded some improvement (Kim et al., 2024; Lee et al., 2024b).

The failure of RL in ARC may be due to several factors: the action space must match an appropriate level of abstraction; reward-shaping is challenging to design; and current RL algorithms may be insufficient for the extreme few-shot generalization required.

Alternatively, it may be that the sequential decision-making framing is misaligned with ARC's structure, since many transformations are easier expressed neurally, or as reusable programs rather than as step-by-step construction procedures, which may result in grid-instance-specific action-sequences. Along similar lines, RL was also applied in neural-guided program search (see 5.1.3). We expect that RL methodology might find more use in ARC-v3, an interactive setting.

# 7 ARC Data Ecosystem

The original ARC-v1 benchmark provides 400 training tasks and 400 validation (public eval) tasks, with typically 2–4 examples each. Researchers have since created complementary resources that provide additional training data, enable more granular evaluation, and support faster prototyping of new approaches.

These auxiliary datasets serve several distinct purposes: synthetic datasets help support data-hungry learning methods; specialized datasets focus on specific reasoning capabilities or settings; and human-centered datasets provide data of how humans see and solve ARC tasks. Together, these resources have influenced the trajectory of ARC research, enabling methodological diversity that would not be possible with the original benchmark alone. For instance, the natural-language task descriptions of H-ARC (LeGris et al., 2024) and BARC (Li et al., 2024a) supported the synthetic data generation pipeline in the 2025 prize-winning entry Sorokin & Puget (2025).

## 7.1 Synthetic Datasets and Extensions

Several works have addressed training data scarcity of ARC-v1 through augmentation and synthetic data generation. Notably Hodel (2024a) introduced **RE-ARC** (Reverse Engineering ARC), a comprehensive procedural example generation framework for ARC, where for each of the 400 training tasks, a generator paired with a verifier (implemented in Python with use of ARC-DSL) mimics the logic of the original task, enabling sampling a vast number of distinct input-output grid pairs. RE-ARC enables systematic study of within-task generalization and sample efficiency, and supports curricula or difficulty modulation. It has been adopted for training by many methods (Bonnet & Macfarlane, 2024; Franzen et al., 2024; Li et al., 2024b).

Li et al. (2024a) introduced **BARC** (Bootstrapping-ARC), consisting of ARC-Heavy and ARC-Potpourri, with 200k and 400k tasks, respectively, synthesized by GPT4, based on 160 human-designed seed tasks (with code and task descriptions). Each task is specified by Python code, with functions to first generate an input grid and then to transform an input grid to an output grid.

Other, simpler data multiplication efforts include **AugARC**, which augments by rotation, mirroring, and permuting examples' order (Bikov et al., 2025), and **Sort-of-ARC** with programmatically generated tasks allowing only 16 shape types (Assouel et al., 2022).

## 7.2 Specialized Datasets

Several subsets isolate specific aspects of ARC for targeted evaluation or simplified prototyping. Xu et al. (2024) introduced **1D-ARC**, restricting tasks to one dimension. Though originally introduced to evaluate LLMs, 1D-ARC was adopted to evaluate hard-to-scale induction logic programming (ILP) methods (Cropper & Cerna, 2025; Hocquette et al., 2024), as well as novel methods not specifically designed for ARC (Wang et al., 2025b; Yu et al., 2025). Moskvichev et al. (2023) created **ConceptARC**, organizing 160 tasks (16 concepts × 10 tasks) around specific capabilities like counting or ordering. Concept-based evaluation allows to reveal lack of generalization with respect to concept variations, where IID accuracy would have been uninformative (Odouard & Mitchell, 2022).

Other contributions include **Object-ARC** (Xu et al., 2023; Lei et al., 2024; Barke et al., 2024), a subset of object-centric tasks from ARC-v1; **MC-LARC** (Shin et al., 2024), formulating tasks as multiple-choice questions over natural-language rule descriptions; **ARAOC** (Wu et al., 2025), targeting six 'atomic operations'; **Compositional-ARC** (Mondorf et al., 2025), a dataset for evaluating systematic generalization of geometric compositions; and **ARCLE** (Lee et al., 2024a), a reinforcement learning environment for ARC.

## 7.3 Human-Centered Datasets

Several datasets offer ARC-related data provided by humans: natural-language descriptions and action traces of human solutions (**H-ARC**; LeGris et al., 2024), instructions produced by humans guiding others to solve tasks through language alone (**LARC**; Acquaviva et al., 2022), recursive decompositions from humans solving tasks programmatically (**DARC**; Huang et al., 2023).

Some research recorded human performance specifically on easy ARC-like tasks. **Mini-ARC** (Kim et al., 2022) includes 5×5 grid tasks and human solution traces, while **KidsARC** (Opiełka et al., 2024) features 3×3 and 5×5 grid tasks and data on children's errors.

Designing tasks, as opposed to solving tasks, is not as well documented. BARC (Li et al., 2024a) (see §7.1), includes human designed tasks, with code, and rule descriptions as comments.

# 8 Discussion

In the six years since its introduction, the Abstraction and Reasoning Corpus has catalyzed a remarkable diversity of approaches and remains challenging without large-scale pretraining or test-time compute. This survey mapped the landscape of methods applied to ARC-v1, from symbolic program synthesis to test-time–adapted transformers, highlighting both current progress and remaining challenges.

## 8.1 State of the Art

**Overview**  Up until 2024, brute-force DSL solvers remained surprisingly competitive, with their best accuracies in the 20–40% range (Wind, 2020; Chollet et al., 2025a). Advanced neurosymbolic solvers achieved limited success, rarely exceeding 20% (Butt et al., 2024; Batorski et al., 2025). The top-scoring solvers to date involve either fine-tuned transformers with test-time tuning, in the 50–60% accuracy range (Franzen et al., 2024; Barbadillo, 2024a; Cole & Osman, 2025), or LLM chain-of-thought sampling (Greenblatt, 2024), reaching accuracy as high as 90% (GPT-5.2 on semi-private evaluation). Evolutionary methods building on top of LLM sampling also provided additional gains, reaching as high as 80% (Berman, 2025).

**Highlights**  ARC motivated development and study of test-time adaptation, for instance, as test-time fine-tuning (Akyürek et al., 2024) or evolutionary test-time compute (Pourcel et al., 2025), and the introduction of novel architectural proposals, inspired from cognitive science (Wang et al., 2025a; Joffe & Eliasmith, 2025), or grounded in information theory (Liao & Gu, 2025). We expect further similar advances in the future; in test-time tuning, like SIFT (Hübotter et al., 2025), and in recurrent architecture design, like TRM (Jolicoeur-Martineau, 2025). Further, we anticipate that ARC-v1, although nearly saturated by LLMs, will continue to be a testbed for neurosymbolic programming (Chaudhuri et al., 2021). We expect ARC-v2 in particular to serve as testing ground for application-level refinements that build on top of frontier commercial LLMs (Greenblatt, 2024; Berman, 2025; Poetiq, 2025), which can increase accuracy at the expense of additional compute. The importance of visual priors had been emphasized recently (Hu et al., 2025; Acuaviva et al., 2025), and we anticipate visual pretraining on real-world data have a greater effect on ARC methodology.

**Many Forms of Test-Time Adaptation**  In ARC, test-time adaptation is a necessity. Thus it proves instructive to observe how solvers across the different categories adapt to the context of a given task. These strategies can be conceptually unified as 'iterative refinement'. Inductive program synthesis searches for a fitting program, iteratively eliminating portions of search space. Chain-of-thought and evolutionary methods refine natural-language rule descriptions, or programs in general-purpose languages. Test-time training in transductive solvers refines the network parameters, thus gradually adapting a high-dimensional curve in its latent space (Knoop, 2025). Some transductive solvers like HRM and TRM (Wang et al., 2025a; Jolicoeur-Martineau, 2025) iteratively refine their predictions throughout the forward-pass.

**LLM Learning**  We concur with the analysis by Greenblatt (2024) in that LLMs perform some degree of useful learning in-context, which at least allows them to write correct Python programs with some non-negligible probability, based on few-shot reasoning traces. However this ability is still very weak when compared to humans, and requires much test-time sampling to elicit. Evolutionary methods provide one particular structured framework for LLM sampling (Berman, 2024; Pourcel et al., 2025).

**The Bitter Lesson**  In line with Sutton's 'Bitter Lesson', we observe that more sophisticated symbolic and neurosymbolic methods were largely overshadowed by the more flexible general-purpose methods better suited for computational scaling, such as transformers fine-tuned to ARC task data, or LLMs generating

programs or reasoning traces en masse. In ARC, this is applicable in two aspects, in pre-test-time preparation (design or training, or both), and test-time adaptation (search, fine-tuning).

However, it does not seem fair to compare the intelligence of certain systems, for instance of CompressARC and LLMs, solely based on their position on the accuracy–cost Pareto frontier, given their substantial differences in prior knowledge and pretraining. If the pretraining costs were included in the computation of cost-per-task, [5] the accuracy–cost Pareto frontier would change substantially from its current state, occupied by small and medium-sized models, such as CompressARC and NVARC. In fact, Chollet's definition (see §1) emphasizes intelligence as skill-acquisition efficiency *with respect to priors*. This raises a broader question of how benchmark performance should be interpreted when systems differ substantially in their sources of prior knowledge.

The increasing diversity of ARC approaches suggests a need to periodically revisit what benchmark performance is intended to measure and whether current evaluation practices remain aligned with that goal. While accuracy was initially the dominant focus, the community has increasingly paid attention to computational cost as an additional axis of comparison. This evolution raises the possibility that other factors may also warrant consideration when comparing solvers.

We consider data exposure to be one promising candidate and outline it as one possible example below, with the goal of illustrating how alternative evaluation perspectives can lead to different conclusions and valuable new insights. As one possible framework, we outline multiple 'data-exposure classes' (by analogy with weight classes in boxing), based on the extent and type of data exposure in (pre)training prior to evaluation:

1. **No pretraining:** Not trained, or trained only on the target task at test-time. Examples are DSL solvers[6] (Wind, 2020), CompressARC (Liao & Gu, 2025), ARC-NCA (Guichard et al., 2025).

2. **Minimal ARC-pretraining:** Trained on not more than 1,000 tasks,[7] being the tasks of the training set, the public evaluation set, and of ConceptARC (Moskvichev et al., 2023). An example is HRM (Wang et al., 2025a).

3. **Heavy ARC-pretraining:** Trained on large-scale synthetic data, e.g. from RE-ARC (Hodel, 2024a) or BARC (Li et al., 2024a). Examples are ARChitects (Franzen et al., 2025), and NVARC (Sorokin & Puget, 2025).

4. **Pretraining on real-world data:** Trained on general data, such as visual data of the physical world, and anthropocentric natural-language data. Examples include solvers that rely on LLM sampling (Greenblatt, 2024; Berman, 2024) and VDMs (Acuaviva et al., 2025).

Note that this comparison has two axes: (1) amount of training data; and (2) its specificity, ranging from ARC-domain-specific to general real-world data. On the opposite ends of these spectra lie two agendas for engineering and research:

1. **Minimalist:** How do we build systems that use minimal data to generalize to solve unseen tasks?

2. **Scale-driven:** How do we make large-scale general-purpose systems learn to solve unseen tasks?

For the time being, intuitions of the Bitter Lesson are contradicted in the no-pretraining class. Concretely, the best-scoring solver is a DSL (40% on private eval; Chollet et al., 2025a), whereas the best neural solver to date obtained 20% on public eval (Liao & Gu, 2025).

Whether such distinctions are ultimately useful, and how they should be formalized, remains an open question.

---

[5] It is true that pretraining costs are amortized in practice—however, ARC only has a small set of evaluation tasks.

[6] Here we do not treat DSL design and engineering process as pretraining, though such an alternative view is viable. However, data exposure from design is hard or impossible to characterize or quantify.

[7] An alternative, stricter limit could be 400, of the official training set.

### 8.2 ARC-v1 as an Intelligence Benchmark

Being able to solve pixel-art IQ-test-like puzzles is not a necessary nor sufficient condition for pragmatic forms of intelligence. The representations in ARC are clean and simplistic. In real-world environments, a major component of problem-solving is selecting the suitable representations.

**Goodhart's Law**   Despite the aim of measuring skill-acquisition ability (where skills roughly correspond to distinct tasks), ARC itself still constitutes a domain of particular specific skills of its own, following a predictable format and recurring types of content. As such, and as a target of competitive incentives and optimization, it has become a less reliable benchmark of intelligence over time. This is in line with Goodhart's law: "when a measure becomes a target, it ceases to be a good measure". The ARC Prize leadership believes this "overfitting" effect is taking place in both ARC-v1 and ARC-v2 (Knoop, 2025).

**Control for Priors**   The original working definition by Chollet (see §1) centers around information-theoretic skill-acquisition efficiency with respect to priors.[8] However, in practice a control for the degree of prior knowledge is hard to implement, without training every solver under evaluation from scratch. In fact, the theoretical formal intelligence measure by Chollet (2019, pp. 38–39) is incomputable, as it involves Kolmogorov complexity.

Due to its popularity over the years, coupled with Goodhart's law, ARC-v1 has somewhat eroded in its capacity to capture developer-aware generalization—which is "the ability of a system, either learning or static, to handle situations that neither the system nor the developer of the system have encountered before" (Chollet, 2019, p. 10). Pfister & Jud (2025) present a compelling critique of ARC-v1 as a benchmark for general intelligence, referring to its fixed format, absence of exploration, and possibility of unrestricted trialing in pretraining.

**Programmatic Task Generation**   Chollet (2019, pp. 55–56) foresaw the limitations of static evaluation tasks, and proposed a solution to programmatically generate new tasks in an open-ended manner. Since high accuracies have been reached by LLMs (85% at $0.17 by Gemini 3 Flash), and we expect inference costs to decrease in the long term, it is reasonable to begin development of this direction now. The problem of generating valid synthetic ARC tasks was not yet systematically studied, since it is in general inherently harder, as validating a task includes ensuring it is solvable. Unless the synthetic task is solvable by construction, one must have a functional ARC task solver in order to evaluate an ARC task generator's quality. We expect that insights from generative–adversarial learning (Goodfellow et al., 2020) open-ended systems (Wang et al., 2019), and teacher–student curriculum learning (Matiisen et al., 2020) will prove helpful in addressing the ARC task generation problem.

**Improvements**   ARC-v3 introduces a dimension of time, as well as elements of interactivity and exploration. We anticipate that this more complex format will help address the issues of ARC-v1 and ARC-v2 outlined above. We also outline some suggestions for increasing the benchmarks' research value, based on our observations throughout preparing this survey:

- Though more challenging technically, defining distributions over task instances (as in RE-ARC), and further, defining distributions over tasks types, would enable a more principled evaluation of a method's ability to adapt to a novel task, in the sense of "adaptation to unknown unknowns" (Chollet, 2019, p. 11).

- Labeling tasks under concept groups, like in ConceptARC, would allow to evaluate generalization more granularly, and concretize complementarity effects.

- Providing a subcorpus of one-dimensional tasks, like 1D-ARC, would allow to study the effects of space on a method's performance (assuming comparable task complexity).

---

[8]'Efficiency' here is sometimes misunderstood as 'computational efficiency'. The two are related but not the same.

**Transferability of ARC Methods**   An important question for future work is which techniques developed within ARC transfer effectively to broader AI settings, and which remain largely specific to the benchmark. Our taxonomy already distinguishes general techniques imported into ARC from broader machine learning (Section 3.4) from contributions native to the benchmark; the reverse direction — how ideas originating in ARC influence neighboring areas — is harder to assess and beyond our present scope. We hope, however, that the shared vocabulary and structure of our taxonomy prove useful as the field maps these connections.

## 9   Conclusion

ARC-v1 is a unique benchmark involving few-shot visual–spatial reasoning and task meta-learning, that aims to capture the ability of systems to adaptively use priors widely shared by humans from early development. It represents a valuable step towards measuring fluid intelligence in machine learning systems. Despite significant progress since its introduction, ARC-v1 remains challenging for contemporary methods—even large language models without substantial test-time compute struggle with it—underscoring that human-like abstract reasoning remains an open challenge. As the ARC research community transitions to ARC-v2, and begins to confront ARC-v3's interactive setting, understanding past efforts becomes increasingly important.

This survey has systematized research on ARC-v1 since its introduction, reviewing approximately 100 works from a previously fragmented landscape and organizing them into a coherent taxonomy. Our taxonomy distinguishes inductive and transductive approaches, and further structures them by their underlying properties and design choices. Besides this hierarchy, our survey also highlights enabling techniques and contributions—ARC-specific infrastructure (DSLs, auxiliary datasets) as well as general techniques (such as search algorithms, test-time adaptation). Together, this provides a comprehensive framework for understanding the diversity of methods applied to ARC. We hope our work and findings will help researchers develop new solvers for visual few-shot reasoning, devise novel measures for machine intelligence, and measure the impact of ARC on the broader field of AI research.

For additional tools and resources, we refer readers to the ARC Prize Foundation's community hub.[9]

### Acknowledgments

The authors used ASReview for pre-screening papers (ASReview LAB developers, 2025). LLM-based scripts[10] were used for initial classification of the collected sources. The authors used LLMs for writing support.

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

## A    Literature Collection Methodology

References were collected by querying Google Scholar and Semantic Scholar on March 16, 2025, for publications citing Chollet (2019) and matching the query `"arc" OR "arc-agi" OR "abstraction and reasoning corpus"`. From the resulting 358 initial records, we identified approximately 100 relevant works after deduplication and screening.

Our inclusion criteria required that papers: (1) be available in full text in English; and (2) propose computational methods for ARC, discuss their design or evaluation, introduce related datasets or tools, or analyze human performance. Some works from 2025 published after the cutoff date were included selectively.

Throughout our analysis, we considered several key dimensions when examining methods: predictive accuracy, training and inference costs, use of Core Knowledge priors, generalization capacity, applicability outside the ARC domain, open-endedness of capabilities, and explainability. This perspective enables synthesis despite differing experimental conditions, assumptions, and resource constraints.

