# OpenReview forum: "A Survey on the Abstraction and Reasoning Corpus"
_TMLR — Accepted by TMLR_

### Review · Reviewer_9vVi · 2026-03-23

**Summary Of Contributions:**

The survey summarizes methodologies that have been applied to abstraction and reasoning corpus (ARC, version 1) and categorizes them under inductive and transductive branches. The survey mainly focuses on methods that are developed for ARC-v1 while also briefly touching upon those that worked on ARC-v2.

I couldn’t find any other comparable survey on this subject (except one in arXiv submitted on March 6, https://arxiv.org/abs/2603.13372), so I believe this survey would serve as a nice summary of recent progress.

**Additional Comments:**

The paper mentions that LLM-based scripts are used for initial classification of the collected sources. It would be good if you could share these scripts.

**Audience:**

Yes

**Audience Explanation:**

The ARC-v1 is a popular challenge to test generalization of methods, and the survey does a good job at collecting recent work on this subject. I believe the paper would draw considerable attention.

**Broader Impact Concerns:**

As the paper is a survey about recent progress in ARC benchmark, I don't see any immediate ethical implications that could rise from this work. The paper doesn't have a broader impact statement, and it might not need one.

**Claims And Evidence:**

Yes

**Claims Explanation:**

The survey indeed summarizes and categorizes recent progress on the ARC-v1 challenge, and it adds some commentary along with each category, helping the reader with the problem framing.

While it might not be fair to expect what I'll be proposing at this point, I would have enjoyed the survey more if approaches were also compared in their generality as opposed to their specificity to the ARC-v1 challenge. For instance, while a data augmentation method specific to the ARC-v1 challenge might help improve the results, it may not be transferrable to other domains. Adding commentary regarding how generic and transferrable vs. task-specific the considered ARC-v1 techniques are would have taken this survey to another level.

**Requested Changes:**

This is not a requested change but my opinion/recommendation on the survey.

If we take a look the overall landscape when Chollet (2019) came up with this challenge, there were many static datasets, mostly for computer vision, focusing on a single task definition with metrics that are saturated. Just as how the survey mentioned the Goodhart's law, whenever a novel dataset is released, for instance, a Q&A dataset, it was becoming quickly saturated. There was some evidence that many CIFAR-10 classifiers were actually overfit to CIFAR-10 test set (https://arxiv.org/abs/1806.00451). If I remember correctly (and forgive me for not appropriately giving references but I hope you'll understand my intention), Chollet's challenge was somewhat a reaction to all these methodologies. Private test sets, changing rules in each example, these were all design decisions to test the method's generalization capability in a hope to measure intelligence.

The survey does a good job at summarizing recent progress on the benchmark. I believe it would have been even better if there was another dimensionality in the paper that focuses on the generality of these methods, how transferrable they are to other tasks, or how much it helped on advancing the field overall. For instance, TTT might be one such example---something that is not specific to ARC. Though, I realize that this would take at least the same amount of time to prepare this survey. And to be fair, there are indeed some commentary scattered in the paper referencing to the generality of the methods.

Here goes my other comments you might consider taking into account.

“Transductive methods include transformer models … or off-the-shelf large language models prompted likewise, in text for-
mat (Xu et al., 2024).” As some LLMs use ‘reasoning’ as an intermediate representation, could they be assumed to be making inductive reasoning even though they are still producing tokens? For instance, an LLM with the reasoning option might first create some rules, and then try to follow them while solving the puzzles (e.g., Section 5.3.2). Yet, there are still no explicit rules, or rather, the rules are rules up to the point that the transformer takes them into account. It might be worth discussing this point briefly.

Following up on the previous point, I think you could include a remark that says the transductive school expects the rules to be implicitly formed inside a neural net as opposed to defining a grammar over which one can search for rules. The view is that this strategy inherently carries the bias of the rule set (or DSL) designer. You could further respond to this view from the transductive’s side. I can’t exactly think of the response at the moment, but it could be a critic of the neural net’s structure, or sample inefficiency. This discussion transcends the ARC challenge and still is an unanswered point, which would be quite useful for this survey as a comparison of paradigms.

I really liked the data-exposure categorization—it may deserve a section on its own—since that gives an idea of the generality of the method vs. tuning it on the task at hand. You might also consider other criteria (somewhat related to this) such as how task-specific the DSLs are.

Typo in Section 4:
“For simple pattern-oriented tasks, the human solvers seem to find a solution easily and intuitively, …”

---

> ### Author Response · Authors · 2026-04-02
> **Author Response to Reviewer 9vVi**
>
> We thank the reviewer for their interest and constructive feedback. We are glad to hear that the paper does a “good job at collecting recent work on this subject” and that you “believe the paper would draw considerable attention.” We address each point in turn.
>
> **Comparable Survey:** Thank you for pointing out the concurrent survey on arXiv (March 2026, https://arxiv.org/abs/2603.13372). As it appeared after we submitted, we will not include it in the paper.
>
> **Dimensions of Analysis (Transferability and Generality):** In two paragraphs, the reviewer notes how further dimensions of analysis, such as transferability of method components to non-ARC tasks, could further improve the paper. We appreciate this thoughtful suggestion and agree that this would add analytical value, but it is beyond the scope of the present work. As stated in the introduction, our goal is descriptive: to provide a unified taxonomy and shared vocabulary for a field that has grown rapidly but remained fragmented. This is already a significant interpretive step for a young area, and we believe the field benefits from first agreeing on the basics — a shared vocabulary and organizational structure — making deeper comparative judgments and analytical frameworks more meaningful. Adding such an axis now would risk layering evaluative judgements on top before the field has agreed on the basics. This however, does not mean that we avoided any form of discussion going beyond summarization and classification, and, as the reviewer noted, we did add some comments when reasonable: “there are indeed some commentary scattered in the paper referencing the generality of the methods.” We believe that addressing the lack of a taxonomy is a more timely concern. We will mention this as important future work.
>
> **Classification of Reasoning LLMs.** In the sentence which you have cited, we discussed LLMs being applied directly without chain-of-thought prompting or harnesses. We place chain-of-thought, (and correspondingly reasoning/thinking LLMs) under "Soft Induction" (namely subsec. 5.3.2) for the same reasons that you have outlined. We will add more clarity on this point to the manuscript.
>
> **Expectations of the Transductive School.** The reviewer suggests "I think you could include a remark that says the transductive school expects the rules to be implicitly formed inside a neural net, as opposed to defining a grammar over which one can search for rules". However, during the literature review process we have not identified this as a consistent expectation among authors in transductive methods, and thus we do not want to make speculative claims on this question. In our view, some researchers may expect merely a decent level of pattern matching. Nonetheless, we are aware of the emergence of such "inductive rule representations" in neural networks, as the reviewer put it: “This discussion transcends the ARC challenge and still is an unanswered point”. This is interesting, and merits attention in future work.
>
> **Data-Exposure Categorization:** We are glad this categorization resonated with the reviewer. Our intention in Section 8.1 was to prompt the community to think more carefully about what ARC performance actually measures. Initially, primarily the achieved score of a method was explicitly considered, but with time, this showed to be insufficient, and cost-per-task was considered as a metric as well. There is a need for the area to take a step back and consider more proactively and explicitly what the benchmark wants to measure and how we can avoid comparing apples to oranges. The data-exposure categorization is a suggestion for consideration, which we do not claim to be formally precise, nor final. It is motivated by the fact ARC does not currently factor the amount and character of the (pre)training data into its measurements. We agree that this is underexplored in the current literature and will mention it as important future work.
>
> Typo in Section 4: Thank you.
>
> **Sharing LLM Classification Scripts:** The scripts were used to generate custom summaries of the collected source papers, including assigning hashtags from a predefined set (#ns for neurosymbolic etc). However, this initial classification served only as an aide to the final deliberation by the authors, and in fact was largely overridden by the introduction of the taxonomy in its final form. We commit to creating a GitHub repo detailing the script workflow and reference it in the paper.
>
> We hope this clarifies the scope and goals of the paper, and we will incorporate the suggested improvements in the revision. Thank you for your time.

---

> > ### Comment · Reviewer_9vVi · 2026-04-02
> >
> > Thank you for your response. I have no further question.

---

### Review · Reviewer_9ADN · 2026-03-27

**Summary Of Contributions:**

The authors perform a survey of methods applied to the AI benchmark ARC-AGI-1.

Strengths:
- The survey is reasonably comprehensive.
- The survey performs a much more in-depth analysis and exposition of the landscape than the concurrent work of Zinkevich (2025).
- The authors do a good job generally summarizing the research landscape.

Weaknesses:
1. The sub-taxonomy under "inductive methods" (in Figure 3) corresponds to the representation of the synthesized rules (DSLs vs "Open-Ended Language"). I think there are two things that can be improved here:
	1. The standard way to refer to what here is called "Open-ended language" is "General-purpose language" (indeed, this is the term that you use in page 6). Is there a reason why the former term is preferred? If not, I would suggest to stick to the field convention.
	2. The more fundamental issue with the taxonomy is that this presentation format suggests that the search techniques used for "Open-Ended Lang." are disjoint from "DSL". Indeed, otherwise why put "LLM Sampling" under "Open-Ended Lang." but not "DSL"? However, that technique can be applied to both; so a naive read of the taxonomy is misleading. Perhaps fitting everything (search methods, representations, etc.) into a single hierarchy is too challenging.
2. The taxonomy proposes groups of "Enabling methods and contributions" and "General methods", which are a one-size-fits-all groups of concepts. This part of the hierarchy is a missed opportunity to leverage established concepts/terms: data (dataset augmentation, auxiliary datasets), training methods (test-time adaptation, Reinforcement Learning, etc.), model classes (VLMs, graphs), search methods (LLM Sampling, etc., see weakness 1.2). This is not me trying to make the authors adapt to my taste, but rather a suggestion and observation that readers may be able to get more out of Figure 3 if the concepts follow standard terminology and there are no generic groups of concepts.
3. Section 3.3 ("Implications and Trade-offs [between transductive and inductive methods]") could be improved substantially in my opinion; that section makes claims without supporting evidence and contains opaque sentences. For example:
	1. "The distinction between inductive and transductive design has significant implications"; the evidence is a quote that only provides intuition.
	2. "Inductive methods may be better suited to generalize"; no evidence is cited.
	3. "Transductive methods [...] are better suited to reproduce output patterns recognized in training data"; no evidence is cited.
	4. "Even if a neural network, with sufficient capacity and training, might obtain an internal representation which is approximately invariant to input grid variations (challenging as it is), the representation being distributed, it would not be readily separable"; it's unclear what would be separable: is it the rule that would be separable?
4. Section 5 "Advantages" and "Challenges" do not contain *any* citations. The content is all likely true, but a survey paper would serve its readers better if it points them to the relevant literature (e.g., which papers support your characterization of the advantages? which papers empirically found the challenges that you list?).
5. Similarly, Section 6's Advantages and Challenges have no pointers to the relevant literature.
6. (Minor) There are no quantitative insights of the surveyed works. E.g., which approaches are more common? Which approaches show better performance? Are there any patterns? Has the number of papers on the benchmark changed over time? Are there any trends over time (e.g., has any group of approaches been abandoned, has anything been discovered recently)?
7. (Minor) There is no summary table of the surveyed works or any other "birds-eye" view of the literature landscape that lists works explicitly.

**Audience:**

Yes

**Audience Explanation:**

The ARC benchmark has been an important part of the AI landscape for the last few years; most practitioners are familiar with it and a broad range of AI/ML sub-fields have contributed proposed methodologies.

However, both the presentation and content of the survey could be strengthened to further increase the value of the manuscript (see Weaknesses above).

**Broader Impact Concerns:**

I do not believe there are concerns on the ethical implications on the work that merit modifications to the manuscript.

**Claims And Evidence:**

No

**Claims Explanation:**

For the most part the authors stick to summarizing the literature, but there are sections with a notable lack with pointers to the relevant data (see Weaknesses 3, 4 and 5 above).

**Requested Changes:**

The most obvious requested change is to acknowledge the existence of the just-released [ARC-AGI-3](https://arcprize.org/arc-agi/3), which was released almost-certainly after the manuscript's submission. E.g., page two refers to ARC-v3 as being "in development".

Other requested changes:
1. I don't have an issue with the emoji in Figure 1 necessarily, but if you insist on having it and because it's not part of the task, then I suggest the caption should (1) warn readers that the emoji is not part of the task, (2) explain the meaning of the emoji.
2.  I suggest Figure 3 sticks to standard terminology if appropriate (see Weakness 1.1)
3.  I suggest Figure 3 is reworked so that different search strategies are not grouped under different program representations (which can be misleading, see Weakness 1.2). I know visualizing everything in a hierarchy is an inherently difficult problem here, but for example, it could be that "Rule Language" and "Search Strategy" are the two children nodes of "Inductive Methods".
3.  I suggest "Enabling methods" be split into more meaningful groups (see Weakness 2)
4.  Strengthen section 3.3 (see Weakness 3).
5.  Add pointers to relevant literature on sections 5 and 6 (see Weakness 4 and 5)

Typos:
- Page 8: "they human"

---

> ### Author Response · Authors · 2026-04-06
> **Author Response to Reviewer 9ADN (Part 1/2)**
>
> We thank the reviewer for their time and feedback. We are glad to hear that the paper does “a good job generally summarizing the research landscape”. We address each point.
>
> **RC0. ARC-v3**
> We will update to acknowledge the release of ARC-v3.
>
> **RC1. Emoji**
> We will clarify the figure caption with "here the test output is marked by an emoji".
>
> **RC2. General-Purpose Languages**
> We agree that "general-purpose" instead of "open-ended" works better for a broader audience, and will edit the manuscript accordingly. The reason we initially adopted the latter was because of its use by Chollet et al. in the ARC Prize Tech Report.
>
> **RC3 and W1.2. Search Strategies and LLM Sampling**
> Our aim is not to partition methods along a single axis, such as search strategy, but rather to reflect differences that lead to fundamentally distinct perspectives and practical requirements. Search strategies appear throughout many approaches, but their centrality varies — for program synthesis, they are not merely interchangeable hyperparameters but define the character of the approach. Greedy Search, Advanced Discrete Search, and Neural-Guided Search represent three families of methods that, while potentially searching the same program space, have significantly different operational consequences in terms of infrastructure, failure modes, and scalability. We will change "LLM Sampling" to "by LLM Sampling", to help dispel any implication that LLM sampling is confined to general-purpose languages. We will also add a clarification to the figure caption.
> The same principle applies to the distinction between DSL and General-Purpose Language search. While General-Purpose Languages can formally be viewed as a discrete action space, LLM sampling operates over much more abstract steps involving many tokens, leading in practice to an effectively open-ended search space with very different considerations. We note that the distinction is not simply "LLM vs. non-LLM" — Neural-Guided Search also includes LLM-based approaches (Sec. 5.1.3), specifically where the LLM guides search over a DSL action space. The key difference is whether the search operates over a structured DSL or an open-ended space. We will clarify this distinction in the paper.
>
> **W6 (Minor). Quantitative Insights.**
> > There are no quantitative insights of the surveyed works.
> > Which approaches show better performance?
>
> In general, we have not focused on the large-scale aggregation and comparison of solver accuracies,
> in part due to (1) the concerns Chollet expressed over the calibration/informativeness of ARC-v1 [e.g. in Chollet et al 2025 "ARC-AGI-2: ..."]
> and (2) the variation in how research papers reported their results (e.g. training/not on the public validation set, evaluating on special subsets, reporting/not the Kaggle private validation set score etc).
> Thus, we focused more so on the breadth of methods, and comparisons among the most well-known, or the best in the Kaggle competitions.
> We also think the ARC Prize Foundation is most apt to maintain such information, as they do at https://arcprize.org/leaderboard (though mostly for LLM solvers).
>
> > E.g., which approaches are more common?
>
> Although we considered it, to answer this question to the fullest extent would necessitate classifying & counting among both published research works, as well as Kaggle submissions (with somewhat expected different proportions).
> Classification would be labor-intensive, and instead we prefer to compare the best submissions in the competitions over the years (e.g. symbolic DSL methods losing to transformers 2021--2024).
>
> > Are there any trends over time (e.g., has any group of approaches been abandoned, has anything been discovered recently)?
>
> We considered the research landscape to be too sparse to seek such conclusions at the present.
>
> **W7 (Minor). Birds-eye View**
> > There is no summary table of the surveyed works or any other "birds-eye" view of the literature landscape that lists works explicitly.
>
> Our intention was for Figure 3 (the taxonomy tree) to serve this purpose. Adding references directly into the diagram would make it cluttered. Since the subsections corresponding to the taxonomy nodes are reasonably small, we hope that the readers will use the references in the main text if needed.
>
> **Typo in Section 4**
> Thank you.
>
> (Continued in Part 2/2)

---

> ### Author Response · Authors · 2026-04-06
> **Author Response to Reviewer 9ADN (Part 2/2)**
>
> (Continued, see Part 1/2)
>
> **RC4 and W2. Enabling Methods Structure**
> We agree that currently the General Techniques subcategory is a one-size-fits-all group of concepts, meaning it is not future-resistant by nature. Our decision was driven by the concern that any sufficiently general categories would have very few children, reducing the diagram's usefulness to readers. After further consideration, however, we agree that the advantages of a more structured future-resistant grouping outweigh the downsides. We will therefore restructure the General Techniques into subcategories with more general, future-resistant terms: Data Handling, Synthesis & Augmentation; Models & Representations; Training Strategies; Test-Time Adaptation; Search Techniques; and Ensembling. We will also clarify that while these categories provide a more stable organizational basis, the specific entries listed in each class reflect the current state of the field and should be understood as a snapshot — new methods will warrant additional entries or further splits over time.
>
> **RC5, W3, Sec 3.3. Evidence in "Implications and Tradeoffs".**
> > "The distinction between inductive and transductive design has significant implications"; the evidence is a quote that only provides intuition.
>
> The quoted paper [Li et al. 2024] also has experimental evidence for neural models of comparable capacity having highly-complementary performance.
> We will make this clearer.
>
> > "Inductive methods may be better suited to generalize"; no evidence is cited.
>
> In the full quotation, we try to argue from first principles/definitions: "Inductive methods may be better suited to generalize *to variations of the transformation rule within one task, since they isolate the underlying rules from the test input*".
>
> > "Transductive methods [...] are better suited to reproduce output patterns recognized in training data"; no evidence is cited.
>
> As in the inductive case, we argue from first principles: "Transductive methods generally impose fewer restrictions on possible transformations—thus they are [...]".
> We agree that this statement is too vague; specifically we had neural transductive methods in mind, and thus we can refer to the Universal Approximation Thm. Thus we will concretize this passage.
>
> > "Even if a neural network, with sufficient capacity and training, might obtain an internal representation which is approximately invariant to input grid variations (challenging as it is), the representation being distributed, it would not be readily separable"; it's unclear what would be separable: is it the rule that would be separable?
>
> Here *it* was meant to refer to the rule representation. We will make this clearer.
>
> We will also rename this subsection to *"Discussion of the Transductive--Inductive Distinction"*, to better signal our initial intention to convey intuitions on the distinction.
> In our view the statements in this subsection are somewhat necessarily opaque, because we must refrain from making definitive statements comparing inductive/transductive systems, as they are in theory equally expressive:
> "the transductive framework subsumes the inductive one, and vice versa; the two classes are in theory equally expressive" (The theoretical equivalence is also posed by [Li et al 2024]).
> If the two frameworks are equivalent in expressivity, then one cannot make statements of the form "The first is better suited for X than the second", unless we have in mind some other consideration, like how the comparison turns out in practice.
>
> Thus the more intuitional claims we make in Subsec. 3.3 are based on the immediate definitions of transd./ind. frameworks from a design perspective,
> whereas the advantages/challenges we present in Sec. 5 and 6 are based on what is commonly observed among these groups.
>
> **RC6, W4,5, Sec 5,6. Citations in "Advantages", "Challenges".**
> Correspondingly, we agree that the Advantages/Challenges paragraphs lack references to the relevant works, and we will amend this.
>
> We hope this addresses the reviewer's concerns and clarifies our design decisions. We will incorporate the suggested improvements in the revision. Thank you for your time.

---

### Review · Reviewer_HzVr · 2026-04-07

**Summary Of Contributions:**

The paper surveys the Abstraction and Reasoning Corpus (ARC), a few-shot visual program synthesis benchmark for AI systems. Methods spanning some 100 papers or other technical communications are categorized as inductive (aiming to produce a program in some language that returns a correct output for a test input) or transductive (aiming to directly predict output from input). Techniques that enable approaches in either camp are also presented. These include data augmentation and test-time training, among others. Human performance on the ARC as well as the variety of ARC-type datasets that have been used in the literature and in competitions are also presented briefly. The survey ends with a discussion on the role of ARC as a benchmark for “intelligence” and proposals for fairer benchmark of few-short reasoning.

Strengths:
- The survey addresses a timely topic;
- The writing is very clear and the organization of the paper is effective;
- A truly wide range of ARC methods are explored. Despite knowing of some of them, I learned of many more reading this paper.

Weaknesses:
- From a benchmarking standpoint, I came out unsure as to which methods are currently the best. The authors do summarize the SOTA in section 8.1 but because different methods have different computational costs (at inference time, but also at training time where applicable), it is difficult to tease out the Pareto frontier of methods.
- I would’ve liked to see a bit more details in describing some of the key methods. For example, for each of the three subsections on inductive methods (Section 5), a template of how each of them works at a high level would be extremely valuable.

**Audience:**

Yes

**Audience Explanation:**

The ARC is now a staple of AI benchmarks with a large body of specialized methods as well as general methods that apply to it. As such, I anticipate substantial interest from TMLR’s audience.

**Broader Impact Concerns:**

None.

**Claims And Evidence:**

Yes

**Claims Explanation:**

The claims made in this submission are based on the ~100 papers/technical posts surveyed therein. Of the tens of these that I am somewhat closely familiar with, I found the authors’ description/analysis to be faithful and suitably contextualized within the broader field of ARC-related papers. Figure 3 proposes a taxonomy of ARC work based on which the body of the paper is structured. I found the taxonomy to be sensible, accurately representing the surveyed papers as far as I could tell.

**Requested Changes:**

- some references are incomplete/missing the blog URL, arxiv or journal/conference information, e.g.: Ainooson et al., Assouel et al., Bikov et al., Bonnet et al., Butt et al., Chollet (2019), and many more. These need to be cleaned up.
- “(Moskvichev et al., 2023) suggested that inductive priors relevant to ARC”: The citation should be in \citeauthor style, as you do in the preceding sentence.
- “In ARC, the true program corresponds to the one that obeys the principle of parsimony (Occam’s razor)”: citation needed for this statement.
- “Thus, here a transformer effectively as an interpreter for latent programs.” —> “Thus, here a transformer effectively serves as an interpreter for latent programs.”

---

> ### Author Response · Authors · 2026-04-15
> **Author Response to Reviewer HzVr**
>
> Thank you for your time and constructive feedback.
>
> **RC#1. References.** We fixed the missing fields in the references in the updated version of the manuscript.
>
> **RC#2,4. Typos.** Fixed, thank you for noticing.
>
> **RC#3. Occam's razor.** This is a fine point, and in fact, the partial sentence you have quoted does not stand. The full sentence continues with "... the principle of parsimony (Occam’s razor), meaning it represents the simplest possible explanation of the observations (the support data), *as perceived by a human*". Thus we meant Occam's razor not in a pure algorithmic sense, but in a more limited, human-centric one. Chollet [2019, p. 35] discusses the relation of parsimony to generalization:
> > "Occam’s razor principle would seem to suggest that the simplest program that works on the training situations should also be a program that generalizes well. However, generalization describes the capability to deal with future uncertainty, not the capability to compress the behavior that would have been optimal in the past – being prepared for future uncertainty has a cost, which is antagonistic to policy compression".
>
> In the updated manuscript we clarified that we mean parsimony as in the simplest explanation with respect to a human sense for addressing future uncertainty; and added a footnote referencing Chollet's remarks on this.
>
> **W1. Pareto frontier.**
> > From a benchmarking standpoint, I came out unsure as to which methods are currently the best.
>
> We did not identify clear winners overall, as we concluded that the comparisons must be conditioned to prior data exposure; amount & nature of pretraining, as we discuss on page 19.
> It would also be risky for example to name as SOTA specific LLM models that are leading at the time of writing, because of the fast-changing nature of the field; but we refer readers to the arcprize.org/leaderboard for the most up-to-date data.
>
> **W2. Detail on key methods.**
> We appreciate this suggestion, but we made a deliberate decision to prioritize breadth over per-method depth, to avoid overloading the survey.
> > for each of the three subsections on inductive methods (Section 5), a template of how each of them works at a high level would be extremely valuable.
>
> Although unified by representation type (domain-specific or general-purpose language / natural language / latent vector), we did not see or conceive of a higher-level template that the diverse solvers in these groups had in common.

---

### Decision · Action_Editor_vLfc · 2026-05-10

**Recommendation:** Accept as is

**Audience:**

Yes

**Audience Explanation:**

All reviewers agree that the survey is a valuable resource acting as an entry point to modern program synthesis techniques applied to ARC.

**Claims And Evidence:**

Yes

**Claims Explanation:**

The paper provides a literature review about recent progress on ARC.